# Four new species of dragon pseudoscorpions (Pseudoscorpiones: Pseudotyrannochthoniidae: *Spelaeochthonius*) from caves in South Korea revealed by integrative taxonomy

Kyung–Hoon Jeong[1,2], Danilo Harms[3,4,5], Jung-sun Yoo ✪[6], Sora Kim ✪[1,2]*

**1** Department of Agricultural Convergence Technology, Jeonbuk National University, Jeonju, Republic of Korea, **2** Lab of Insect Phylogenetic & Evolution, Jeonbuk National University, Jeonju, Republic of Korea, **3** Museum of Nature Hamburg – Zoology, Institute for the Analysis of Biodiversity Change, Hamburg, Germany, **4** Research Associate, Harry Butler Institute, Murdoch University, Murdoch, Australia, **5** Research Fellow, Department of Zoology and Entomology, University of the Free State, Bloemfontein, South Africa, **6** Wild Animal Quarantine Center, Incheon, Republic of Korea

* skim01@jbnu.ac.kr

## Abstract

Karst research in Korea is still in its infancy and the invertebrate fauna of subterranean systems across the country is poorly known. One of the very diverse lineages in caves across Korea, the pseudoscorpions, are almost undocumented although they represent stunning examples of cave adaptations and troglomorphism. In this study, we provide a phylogenetic hypothesis for the pseudoscorpion *Spelaeochthonius* Morikawa, 1954 (Pseudoscorpiones: Pseudotyrannochthoniidae) in South Korea; a genus that exclusively occurs in caves across China, Japan and the Korean Peninsula. We report seven species of which four are newly described and illustrated based on molecular, distributional and morphological data: *Spelaeochthonius dugigulensis* **sp. nov.**, *S. geumgulensis* **sp. nov.**, *S. magwihalmigulensis* **sp. nov.** and *S. yamigulensis* **sp. nov**. All species are strongly cave-adapted and known from a single cave or karst system only, emphasizing the need to implement conservation strategies for Korean karst systems and their fauna.

## 2. Introduction

Recent studies have shown that South Korea may have more than 1,000 caves of which a significant proportion remains unexplored [1]. These caves encompass limestone cave systems and lava tubes spread across two extensive mountain ranges, the Taebaek Mountain Range and the Sobaek Mountain Range. These mountain ranges have served as refugia during glacial periods and as barriers to speciation in many animal lineages [2–4] but little is still known about subterranean fauna that occurs beneath the surface in caves of different geological age and origin [5–7].

**Data availability statement:** The paper contains the Genbank accession number of every specimen used in this phylogenetic analyses, the explanation of where the DNA is deposited, and the voucher number of each DNA.

**Funding:** This work was supported by a grant from the National Institute of Biological Resources (NIBR), funded by the Ministry of Environment (MOE) of the Republic of Korea (NIBR202502102). There was no additional external funding received for this study.

**Competing interests:** The authors have declared that no competing interests exist.

One of the most diverse animal groups in karstic caves across the Korean Peninsula is pseudoscorpions (Arachnida: Pseudoscorpions) but this fauna has not been studied until recently [6]. In Korea, pseudoscorpions are a "neglected" invertebrate taxon with presently 28 species in 13 genera that belong to eight families [8]. However, a recent preliminary barcoding study has indicated that the species biodiversity in surface habitats must be significantly higher [9]. The subterranean fauna is probably also diverse, but to date, only five pseudoscorpion species have been recorded from caves in South Korea [6,10]. This is in sharp contrast to other countries in the region such as China where many new pseudoscorpion species are described each year, many of which were sampled from caves [11–13].

To date, all troglomorphic (strongly cave-adapted) pseudoscorpions in Korea belong to a single genus and family (Pseudotyrannochthoniidae: *Spelaeochthonius* Morikawa, 1954) that occurs in Korea with three described species: *Spelaeochthonius cheonsooi* You, Yoo, Harvey, and Harms, 2022; *S. dentifer* (Morikawa, 1970); and *S. seungsookae* You, Yoo, Harvey, and Harms, 2022 (Figs 1–2). This genus is generally interesting because it is widespread in eastern Asia and has eleven species in China, Japan and Korea (Fig 1). All species are strongly cave-adapted and

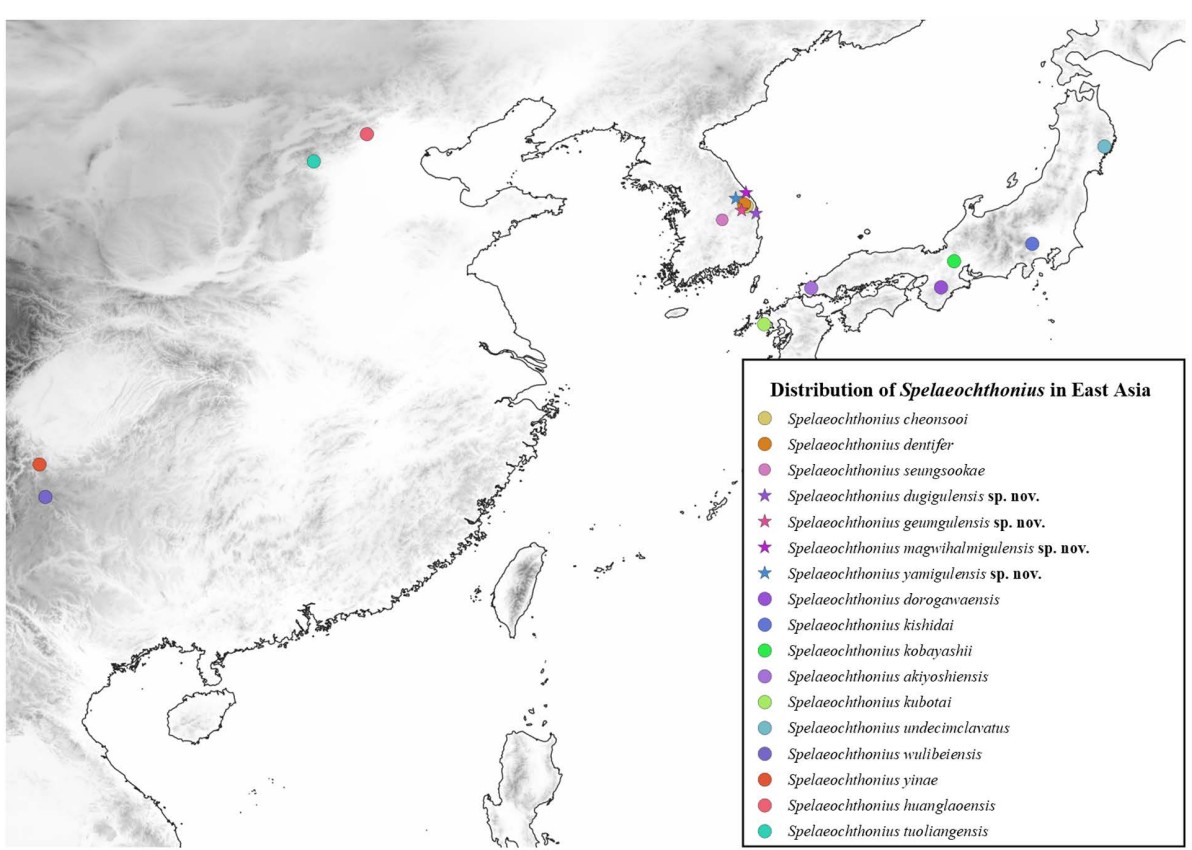

**Fig 1. Distribution of *Spelaeochthonius* species in East Asia.** Distributions of species are followed by [8, this study]. Maps are generated by the combination of USGS GMTED2010 and Natural Earth, which comply with CC BY 4.0.

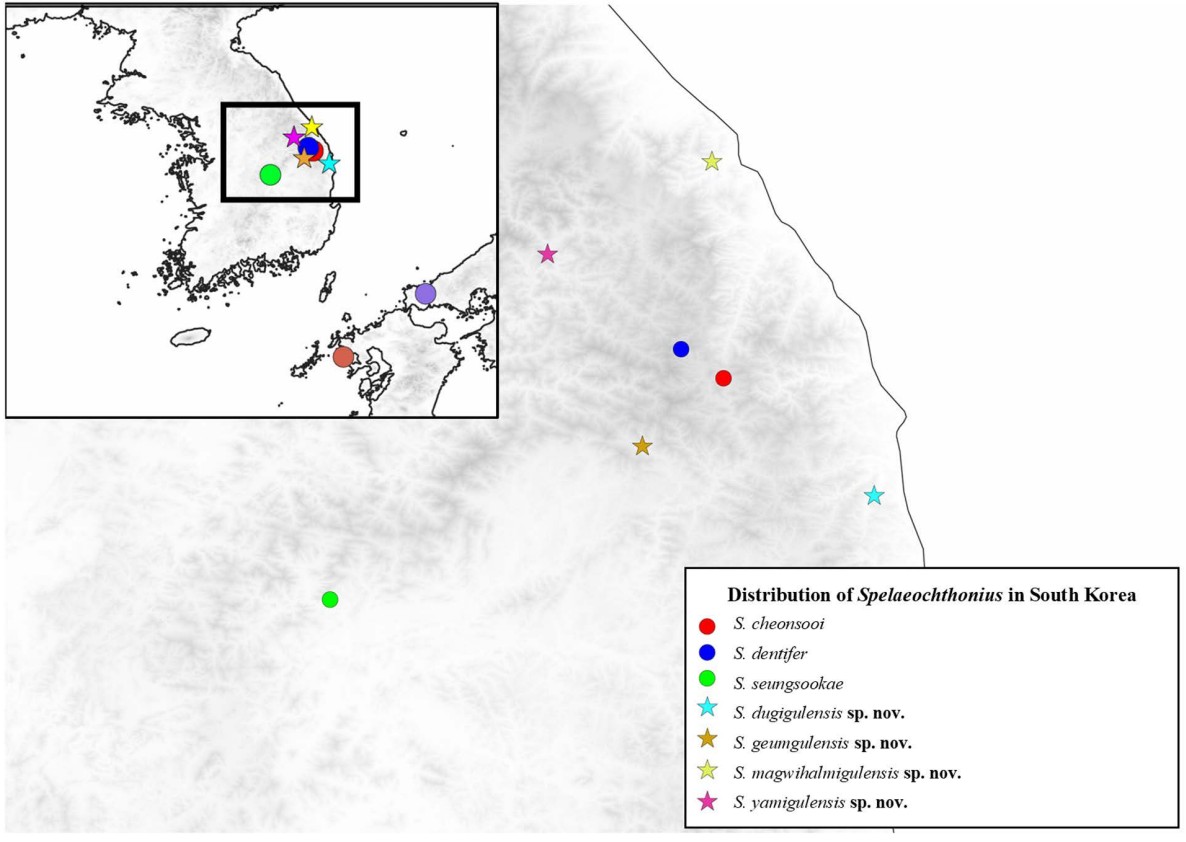

**Fig 2. Distribution of *Spelaeochthonius* species in South Korea.** Distributions of species are followed by [8, this study]. Maps are generated by the combination of USGS GMTED2010 and Natural Earth, which comply with CC BY 4.0.

troglomorphic, meaning that they lack eyes and body pigmentation. The genus is replaced in surface habitats by the related genus *Allochthonius* Chamberlin, 1929 in the same family, and *Spelaeochthonius* might be an ancient relict lineage that was more widespread in the past but is now confined to subterranean refugia. Recent field work in caves across the Korean Peninsula suggests that *Spelaeochthonius* is indeed very diverse in South Korea and shows extreme degrees of regional endemism, but there are also practical problems because species are often rare in their natural habitats and difficult to sample [6].

This paper is part of an ongoing systematic study of pseudoscorpions in Korea [14,15] and adds to our knowledge of *Spelaeochthonius* on the Korean Peninsula by describing four new species from South Korea based on morphological and molecular analyses. All species are single-karst endemics and therefore of conservation relevance. We also provide a preliminary phylogeny for *Spelaeochthonius* using molecular sequences of the Cytochrome Oxidase subunit I (COI) "barcoding" gene, which is commonly used for species delimitation in pseudoscorpions [9,16–18], and an updated distribution map for the genus in eastern Asia (Figs 1–2).

## 3. Materials and methods

### 3.1. Field sampling and morphological identification

All specimens used for taxonomy were collected from four caves located in the Gangwon-Province (GW), Gyeongsangbuk-Province (GB), and Chungcheongbuk-Province (CB) and preserved directly in 70% ethanol (Fig 3). Field surveys were

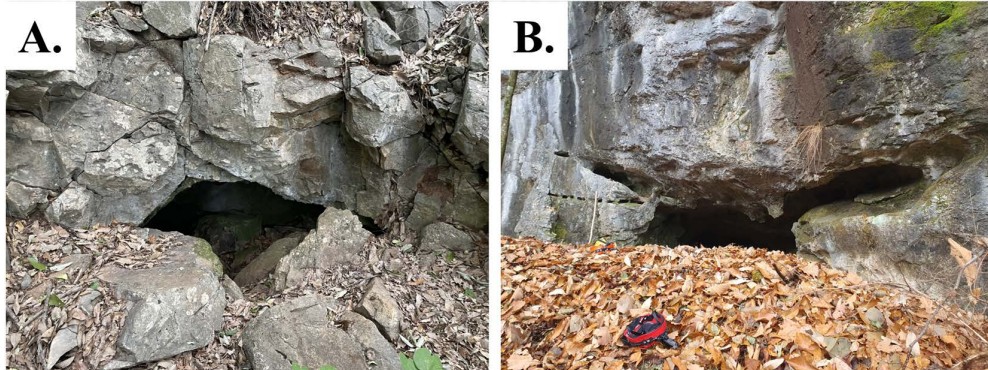

**Fig 3. Images of caves in South Korea.** A. Dugi-gul cave; B. Yami-gul cave.

conducted with the assistance of the Korean Society of Cave Environment or conducted in publicly accessible areas. Identification was conducted using a Leica Z16 APO stereomicroscope, with images captured using a Leica Z16 APO stereomicroscope and Dhyana 400 DC (4M) sCMOS camera (TUCSEN, Fuzhou, China) using Mosaic Analysis Software 2.4. Images were edited using Adobe Photoshop 2024 (Adobe Inc.) and illustrations were drawn using Adobe Illustrator 2024 (Adobe Inc.). The format for the description follows previous papers [14,15]. Distribution maps were created using GMTED2010 and Natural Earth public domain datasets, which comply with CC BY 4.0, and generated using QGIS desktop 3.34.0 (OSGeo). Measurements and terminology follow Chamberlin (1931), Harvey (1991), Judson (2007) and Kolesnikov et al. (2023) [19–22]. Abbreviations for chelal trichobothria: *b* – basal, *sb* – subbasal, *st* – subterminal, *t* – terminal, *ib* – internal basal, *isb* – internal subbasal, *eb* – external basal, *esb* – external subbasal, *it* – internal terminal, *ist* – internal subterminal, *et* – external terminal, *est* – external subterminal, *xs* – duplex trichobothria. Abbreviations for carapacal setae: *Ao* – anterior ocular seta, *Al* – anterolateral seta, *Am* – anteromedial seta, *Ol* – lateral ocular seta, *Om* – medial ocular seta, *Ml* – medial ocular seta, *Il* – intermedialateral seta, *Pl* – posterolateral seta, *Pm* – posteromedial seta. To safeguard biologically sensitive cave habitats, the coordinates of the caves have been rounded to the nearest minute. This measure ensures that the precise locations remain protected from potential disturbances.

### 3.2. Molecular methods and analyses

A total of 14 sequences were used for molecular phylogenetic analyses of the mitochondrial Cytochrome Oxidase subunit I (COI) "barcoding" gene, of which nine were newly generated in this study. Published sequences for *S. cheonsooi*, *S. dentifer*, *S. seungsookae*, *S. kobayashii*, and *Spelaeochthonius* sp. were sourced from NCBI (GenBank accession no. MZ394005–MZ394007, OR290003, OR290018), and eight new sequences were identified, corresponding to *S. dugigulensis* **sp. nov.**, *S. geumgulensis* **sp. nov.**, *S. magwihalmigulensis* **sp. nov.**, and *S. yamigulensis* **sp. nov** (Genbank accession no. PV640640–PV640647). The DNA were provided and deposited at the Laboratory of Insect Phylogenetics and Evolution, Jeonbuk National University (JBNU IPE), Republic of Korea, under the voucher number KHDNA001–008. An additional sequence of *Allochthonius buanensis* Lee, 1982 was used as an outgroup (Genbank accession no. OR290020).

DNA extraction was performed using two to three legs detached from each specimen with the DNeasy Blood & Tissue Kit (QIAGEN, Valencia, CA) and LaboPass™ DNA Purification Kit (Cosmo Genetech Co. Ltd., Seoul, South Korea) following the manufacturer's protocol. PCR amplification for a 650 bp fragment of the COI gene was achieved by using the primer set LCO 1490 (5'-GGTCAACAAATCATCATAAAGATATTGG-3') [23] and HCOoutout (5'-GTAAATATATGRT GDGCTC-3') [24] and under the following conditions: initial denaturation at 94°C for 2 min, followed by 40 cycles of denaturation at 95°C for 30 sec, annealing at 40–45°C for 30 sec, extension at 72°C for 1 min, and a final extension at 72°C

for 5 min. DNA amplification utilized AccuPower PCR Premix (Bioneer, Daejeon, South Korea) and PCR products were examined in 1.2% agarose gels and sequenced at Macrogen, Inc. (Geumcheon–Gu, Seoul, South Korea).

Intra– and interspecific pairwise genetic distance calculations were conducted using MEGA 11.0 and the Kimura 2-parameter model [25]. A phylogenetic analysis was conducted using Maximum Likelihood (ML) methods on the IQ-TREE web-server (http://iqtree.cibiv.univie.ac.at/) using the GTR + G + I substitution model [26]. Tree was drawn and edited using FigTree v.1.4.4 and Adobe Illustrator 2024 (Adobe Inc.).

### 3.3. Nomenclatural acts

The electronic edition of this article conforms to the requirements of the amended International Code of Zoological Nomenclature, and hence the new names contained herein are available under that Code from the electronic edition of this article. This published work and the nomenclatural acts it contains have been registered in ZooBank, the online registration system for the ICZN. The ZooBank LSIDs (Life Science Identifiers) can be resolved and the associated information viewed through any standard web browser by appending the LSID to the prefix "http://zoobank.org/". The LSID for this publication is: urn:lsid:zoobank.org:pub:A04FD752-955E-4B5B-BBE7-D0A293F47C23. The electronic edition of this work was published in a journal with an ISSN, and has been archived and is available from the following digital repositories: PubMed Central, LOCKSS.

## 4. Results

### 4.1. Taxonomy

**Family Pseudotyrannochthoniidae Beier, 1932**

 **Genus *Spelaeochthonius* Morikawa, 1954**

 *Spelaeochthonius* Morikawa 1954: 83–84. Type species: *S. kubotai* Morikawa, 1954

 *Allochthonius* (*Spelaeochthonius*): Morikawa 1960: 104–105

 *Pseudotyrannochthonius* Muchmore 1967: 132

 *Spelaeochthonius* You, Yoo, Harvey and Harms 2022: 135–157

 **Diagnosis.** *Spelaeochthonius* is most similar to *Centrochthonius* Beier, 1931 because both genera share the presence of 16 setae on carapace in most species (Figs 4A, 4C) whereas other pseudotyrannochthoniid genera (e.g., *Allochthonius*) generally have more than 16 setae (Fig 4E). *Spelaeochthonius* differs from *Centrochthonius* by having more than six coxal spines on leg coxa I that are also distally elongated (Figs 4B, 4D) (You *et al.* 2022). *Spelaeochthonius* also differs from *Allochthonius*, which occurs in the same area, by the number of carapacal setae (16 in *Spelaeochthonius*, 14–28 in *Allochthonius*) (Figs 4A, 4E), the form of coxal spines (short and arranged as an oblique line in *Spelaeochthonius*; fan-shaped and on a common tubercle in the genus *Allochthonius*) (Figs 4B, 4F), and the size of the intercoxal tubercle (generally larger in *Allochthonius*) (Figs 4B, 4F).

 **Distribution.** Exclusively found in caves in Japan (six species); China (four species); and South Korea (seven species; but see below).

 ***Spelaeochthonius dugigulensis* Jeong & Harms sp. nov.**

 **urn:lsid:zoobank.org:act:CD314568-4904-4762-9 CD1-9B52354199D6**

 Figs 1–2, 5–6

 **Type material.** Holotype. Male (NIBRIV0000924112 KOREA: Gyeongsangbuk-Province: Uljin-gun, Maehwa-myeon, Dugigul-cave; 36˚53'N 129˚21'E; 25 Sep. 2022; KH Jeong leg.)

 Paratypes. three females (NIBRIV0000924113). Same data as holotype.

 **Etymology.** This species is named after the cave "Dugigul" in Uljin-gun, Gyeongsangbuk-Province, where all specimens were collected.

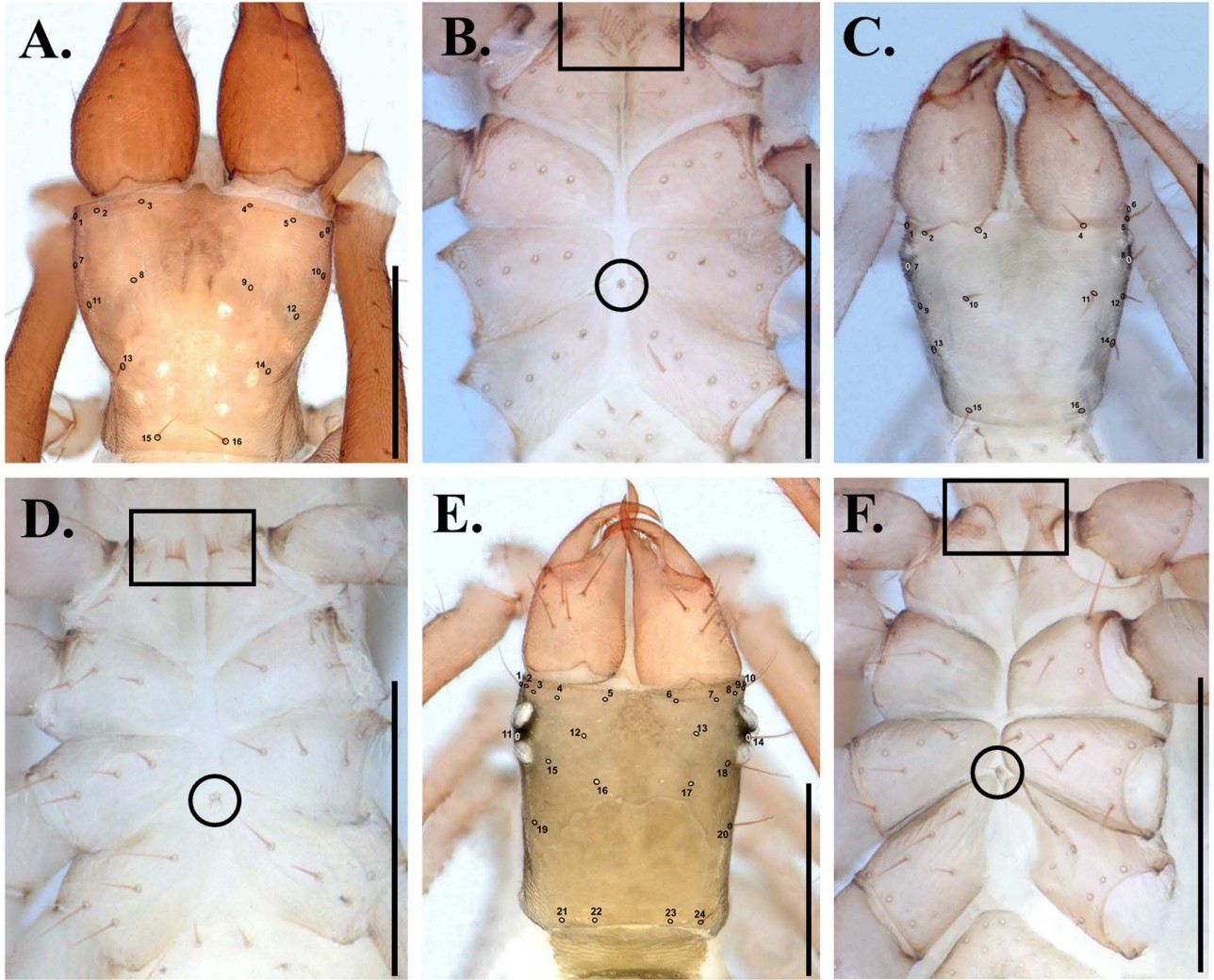

**Fig 4. Carapace and coxa of pseudotyrannochthoniids.** A. Carapace of *Spelaeochthonius*; B. Coxa of *Spelaeochthonius*; C. Carapace of *Centrochthonius*; D. Coxa of *Centrochthonius*; E. Carapace of *Allochthonius*; F. Coxa of *Allochthonius*. Square boxes indicate the position of coxal spines, and circles indictae the position of intercoxal tubercles.

**Diagnosis.** This species is most similar to *Spelaeochthonius akiyoshiensis* from Japan but both species differ in the following characters: L/W ratio of chela (> 6x in *S. dugigulensis* **sp. nov.** and <6x *in S. akiyoshiensis*), number of setae on cheliceral palm (five vs. six), and the number of marginal teeth in chelal fingers (>30 vs. < 30).

**Description. Male, adult (holotype**, Figs 5A–5B)

Colour. Uniformly reddish orange.

Chelicera (Fig 6D). Palm slightly granulate and with five setae, one seta on movable finger; nine marginal teeth on movable finger, eight on fixed finger; two large teeth on the middle of fixed finger; serrula exterior with 25 blades, terminal blades distinctly bigger than basal blades; rallum with 11 pinnate blades; one dorsal lyrifissure.

Pedipalp (Figs 6E–6F). Trochanter 1.86 times, femur 6.75 times, patella 3.5 times, chela 6.13 times, hand 2.3 times longer than broad, movable finger 1.67 times longer than the hand. Fixed finger with eight trichobothria, movable finger with four trichobothria; *isb* and *ib* basally positioned on the dorsum of fixed finger; *eb*, *ist*, and *esb* grouped, *ist* clearly

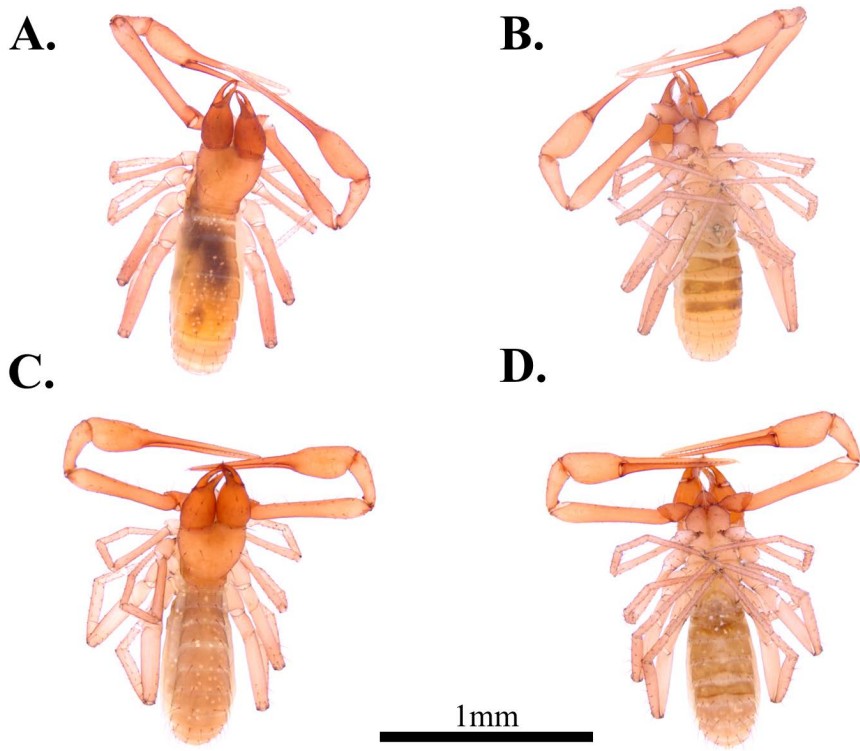

**Fig 5. Habitus of *Spelaeochthonius dugigulensis* sp. nov.** A. Holotype male, dorsal view; B. Holotype male, ventral view; C. Paratype female, dorsal view; D. Paratype female, ventral view. Scale bar: 1 mm.

closer to *esb*; *est* and *it* positioned on the middle of fixed finger; *et* positioned next to *xs*, about two areolar apart; *xs* located terminally on the fixed finger; *sb*, *b*, and *t* separately situated from *st*; *st* positioned basally on the movable finger. Fixed finger with 34, movable finger with 36 spaced, apical marginal teeth.

Cephalothorax (Figs 6A–6C). Carapace 0.99 times longer than broad, subquadrate, eyeless; epistome pronounced and triangular; 16 setae on the carapace, three pairs of seate on the anterior margin (*Ao*, *Al*, *Am*), a pair of setae on the lateral margin (*Ol*), three setae pairs on the medial margin (*Om*, *Ml*, *Il*), one setae pair on the posterior margin (*Pm*). Two pairs of anterior lyrifissures, a pair of posterior lyrifissures present. Two acuminate setae on the manducatory process, maxilla with three setae; coxal chaetotaxy 7: 4: 5: 4; coxal spines with nine blades on a low mound; each spine tripartite; bisetose intercoxal tubercle present between coxa III and IV.

Abdomen. Pleural membrane granulate; tergites undivided; tergal chaetotaxy 2: 2: 4: 4: 6: 6: 7: 7: 5: 4: 0: 0; sternite III–V divided; sternal chaetotaxy 12: 21: 12: 10: 10: 9: 10: 8: 5: 2: 2.

Legs. Leg I: trochanter 1.56 times, femur 7.1 times, patella 5.63 times, tibia 6.17 times, tarsus 10 times longer than broad. Leg IV: trochanter 1.71 times, femur+patella 4.08 times, tibia 8 times, metatarsus 3.56 times, tarsus 10.78 times longer than broad; arolium undivided; pseudotactile seta present on leg IV tarsus and metatarsus.

Male genitalia. 12 setae on the sternite II, six setae on the anterior margin of the genital opening area, eight setae in the genital opening area, 21 setae on the sternite III, and 11 setae on the posterior margin of the genital opening area. Sternite II with two lyrifissures; sternite III with four lyrifissures.

Dimensions (in mm). Body length 2.17. Pedipalp: trochanter 0.39/0.21, femur 1.35/0.20, patella 0.70/0.20, chela 1.84/0.30, hand 0.69/0.30, movable finger 1.15. Chelicera: total 0.64/0.30, hand 0.30/0.30, movable finger 0.34.

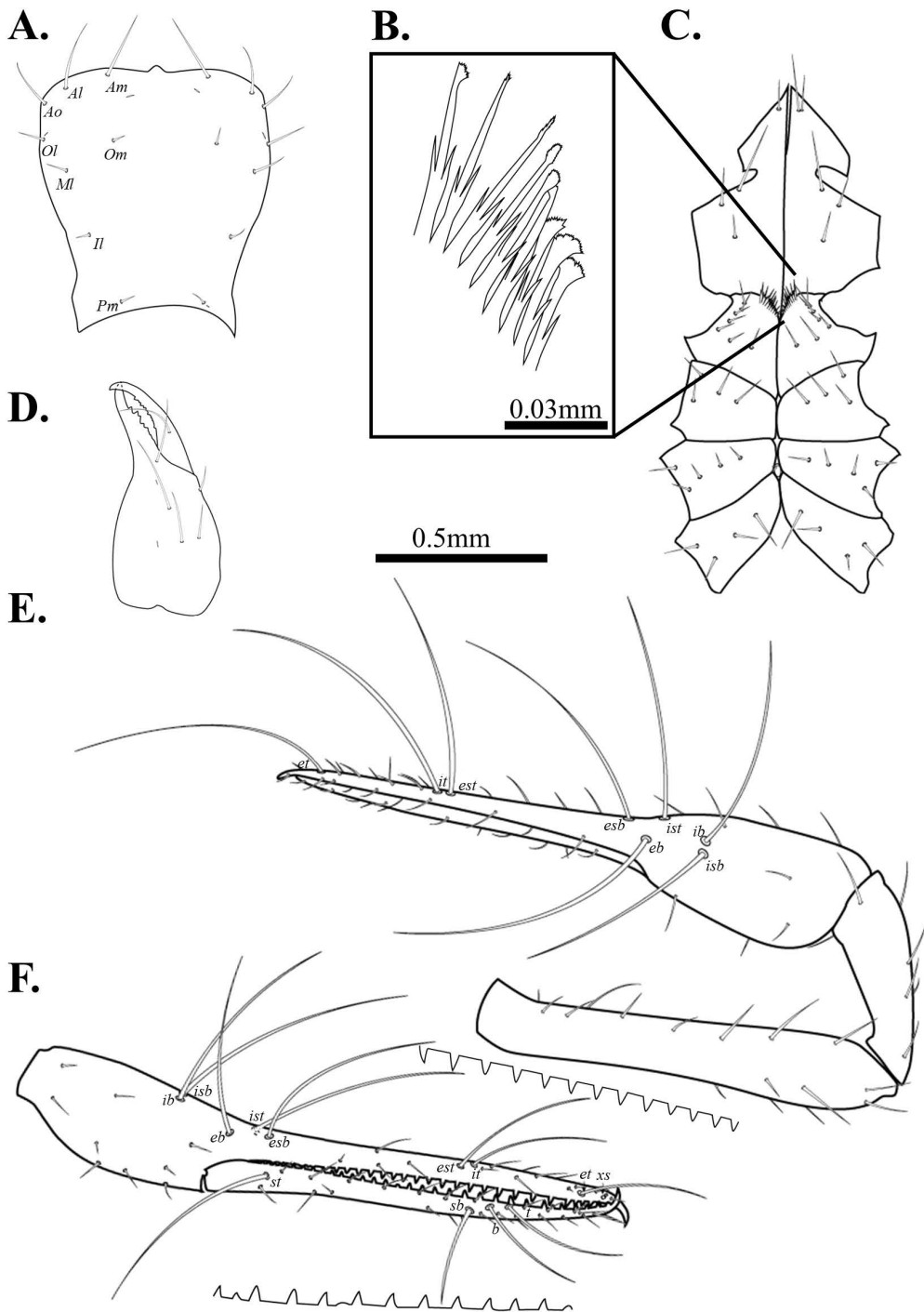

**Fig 6. Drawings of *Spelaeochthonius dugigulensis* sp. nov.** A. Carapace, dorsal view; B. Coxal spines; C. Coxa; D. Chelicera, dorsal view; E. Right pedipalp, dorsal view; F. Right chela, lateral view. Scale bars: 0.5 mm (A, C–F); 0.05 mm (B).

Leg I: trochanter 0.25/0.16, femur 0.71/0.10, patella 0.45/0.08, tibia 0.37/0.06, tarsus 0.80/0.08. Leg IV: trochanter 0.29/0.17, femur+patella 0.98/0.24, tibia 0.80/0.10, metatarsus 0.32/0.09, tarsus 0.97/0.09.

**Variation (females, paratypes, Figs 5C–5D).** Same as the holotype.

Pedipalp. Trochanter 1.44–1.79 times, femur 5.05–5.70 times, patella 2.22–2.79 times, chela 5.77–5.80 times, hand 1.91–1.93 times longer than broad, movable finger 2.01–2.02 times longer than the hand. Movable finger with 30–32, fixed finger with 28 marginal teeth.

Cephalothorax. Carapace 0.95–1.10 times longer than broad; with 16 setae; six setae on the anterior margin, and two setae on the posterior margin. Coxal chaetotaxy: 6–7: 4–5: 4: 4–6; coxa I with nine spines.

Abdomen. Tergal chaetotaxy 2: 2: 4: 6: 6: 6–7: 6–7: 7–8: 5–7: 4–5: 0: 0. Sternal chaetotaxy 10–12: 12–14: 12: 10–12: 8–11: 9–12: 8–10: 6–7: 5: 2: 2.

Legs. Leg I: trochanter 1.46–1.52 times, femur 6.58–6.68 times, patella 4.04–4.94 times, tibia 3.74–4.34 times, tarsus 7.11–8.07 times longer than broad; leg IV: trochanter 1.37–1.72 times, femur+patella 3.50–3.97 times, tibia 5.69–6.05 times, metatarsus 2.79–2.94 times, tarsus 6.28–9.50 times longer than broad.

Genitalia. 10–12 setae on the genital opening area; 12–14 setae on sternite III.

Dimension (in mm). Body length 2.05–2.07. Pedipalp: trochanter 0.30–0.32/0.18–0.21, femur 0.95–1.13/0.19–0.20, patella 0.45–0.59/0.20–0.21, chela 1.55–1.66/0.27–0.29, hand 0.51–0.55/0.27–0.29, movable finger 1.04–1.11. Chelicera: total 0.68–0.72/0.31–0.32, hand 0.37–0.40/0.31–0.32, movable finger 0.38–0.40. Leg I: trochanter 0.21–0.22/0.14–0.15, femur 0.55–0.58/0.08–0.09, patella 0.32–0.38/0.08, tibia 0.24–0.28/0.06–0.07, tarsus 0.42–0.62/0.06–0.08. Leg IV: trochanter 0.25–0.28/0.16–0.18, femur+patella 0.82–0.84/0.21–0.24, tibia 0.64–0.68/0.11, metatarsus 0.21–0.22/0.07–0.08, tarsus 0.40–0.59/0.06.

***Spelaeochthonius geumgulensis* Jeong & Harms sp. nov.**

**urn:lsid:zoobank.org:act:CD3C1C6F-DF94-440F-9990-9A50BDD26B77**

Figs 1–2, 7–8

**Type material.** Holotype. Male (NIBRIV0000924114 KOREA; Chungcheongbuk-Province, Danyang-gun, Danyang-eup, Dodam-ri 47, Geum-cave; 36˚59'N, 128˚51'E; 19 July. 2022; J.H. Oh leg.

Paratype. One female (NIBRIV0000924115), same data as holotype

**Etymology.** This species is named after the "Geumgul" cave in Danyang–gun, Chungcheongbuk–Province, where all specimens were collected.

**Diagnosis.** This species is most similar to *Spelaeochthonius dentifer* but differs in the length of femur IV (0.98–1.07 mm in *S. geumgulensis* **sp. nov.** vs. 0.97 mm in *S. dentifer*), the count of marginal teeth on the fixed chelal fingers (29–33 vs. 34), and ratios of leg IV tarsus (< 10 times longer than broad; > 10 times longer than broad).

**Description. Male, adult (holotype. Figs 7A–7B)**

Colour. Uniformly pale–orange, legs paler than the body.

Chelicera (Fig 8D). Cheliceral palm granulate and with six setae, movable finger with one seta in medial position; 15 marginal teeth on the movable finger, seven on the fixed finger; one big tooth on the fixed finger; rallum with 12 blades; serrula exterior with 25 blades, terminal blades bigger than the basal blades; one dorsal lyrifissure.

Pedipalp (Figs 8E–8F). Trochanter 1.73 times, femur 6.15 times, patella 2.77 times, chela (with pedicel) 6.86 times, hand 2.58 times longer than broad, movable finger 1.66 times longer than the hand. Fixed finger with eight trichobothria, movable finger with four trichobothria; *isb* and *ib* situated basally on the dorsum of the fixed finger; *eb*, *ist*, and *esb* grouped, *ist* slightly closer to *esb*; *est* and *it* positioned on the middle of the fixed finger, *it* closely located to *et*; *et* positioned on the middle of *it* and *xs*; *xs* located terminally on the fixed finger; *sb*, *b*, and *t* separately situated from *st*; *st* positioned basally on the movable finger. Both fingers with long and triangular teeth, spaced together; fixed finger with 29 marginal teeth, 30 on the movable finger.

Cephalothorax (Figs 8A–8C). Carapace 1.09 times longer than the broad and subquadrate, posterior margin narrower than the anterior margin; all setae short and acuminate; 16 setae on the carapace, three setae pairs on the anterior margin (*Ao*, *Al*, *Am*), one setae pair on the lateral margin (*Ol*), three setae pairs on the medial margin (*Om*, *Ml*, *Il*), one setae

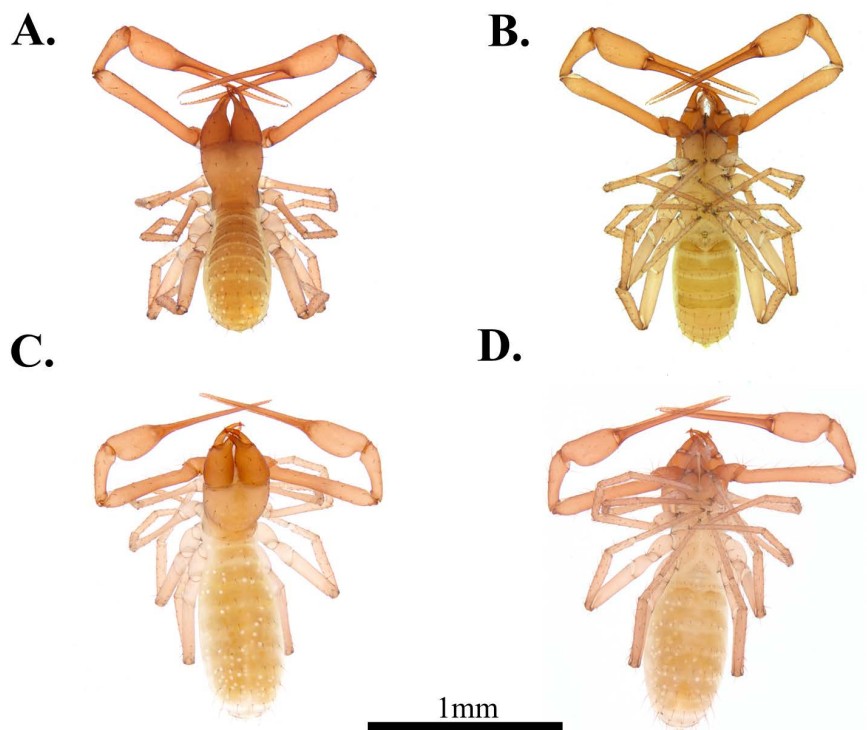

**Fig 7. Habitus of *Spelaeochthonius geumgulensis* sp. nov.** A. Male, dorsal view; B. Male, ventral view; C. Female, dorsal view; D. Female, ventral view. Scale bar: 1 mm.

pair on the posterior margin (*Pm*); without eye; epistome small and triangular. Two pairs of anterior lyrifissures, one pair of posterior lyriffisure present. A manducatory process with one acuminate seta; four setae on the maxilla; coxal chaetotaxy 6: 4: 4: 4. Nine spines on the coxa I; each spine with tripartite shape. Intercoxal tubercle distinctly present between coxa III and IV, two setae present.

Abdomen. Pleural membrane granulate. Tergites undivided; tergal chaetotaxy 2: 4: 4: 4: 6: 6: 6: 7: 6: 4: 0: 0. Sternite III, and IV divided; sternal chaetotaxy 13: 22: 16: 12: 12: 10: 10: 7: 6: 2: 2.

Legs. Leg I: trochanter 1.46 times, femur 7.95 times, patella 6.23 times, tibia 4.58 times, tarsus 10.76 times longer than broad. Leg IV: trochanter 1.17 times, femur+patella 4.26 times, tibia 6.22 times, metatarsus 2.66 times, tarsus 8.59 times longer than broad. Arolium undivided.

Male genitalia. 13 setae on the sternite II, six setae situated on the anterior margin of the genital opening area; 22 setae on the sternite III, five each positioned on each side of the opening area. One pair of lyriffisure on sternite II, one pair of lyriffisure on sternite III.

Dimensions (in mm). Body length 2.13. Pedipalp: trochanter 0.37/0.22, femur 1.23/0.20, patella 0.60/0.22, chela (with pedicel) 1.86/0.27, movable finger 1.16, hand 0.70/0.27. Chelicera: total 0.69/0.30, movable finger 0.36, hand 0.63/0.30. Carapace 0.69/0.64. Leg I: trochanter 0.25/0.17, femur 0.74/0.09, patella 0.44/0.07, tibia 0.31/0.07, tarsus 0.74/0.07. Leg IV: trochanter 0.29/0.25, femur+patella 0.98/0.23, tibia 0.74/0.12, metatarsus 0.24/0.09, tarsus 0.57/0.07.

**Variation (female, paratype, Figs. 7C–7D).** Pedipalp. Trochanter 1.70 times, femur 5.66 times, patella 2.92 times, chela (with pedicel) 6.91 times, hand 2.17 times longer than broad, movable finger 1.83 times longer than hand. Movable finger with 32, fixed finger with 31 marginal teeth.

Cephalothorax. Carapace 1.13 times longer than broad, with 16 setae.

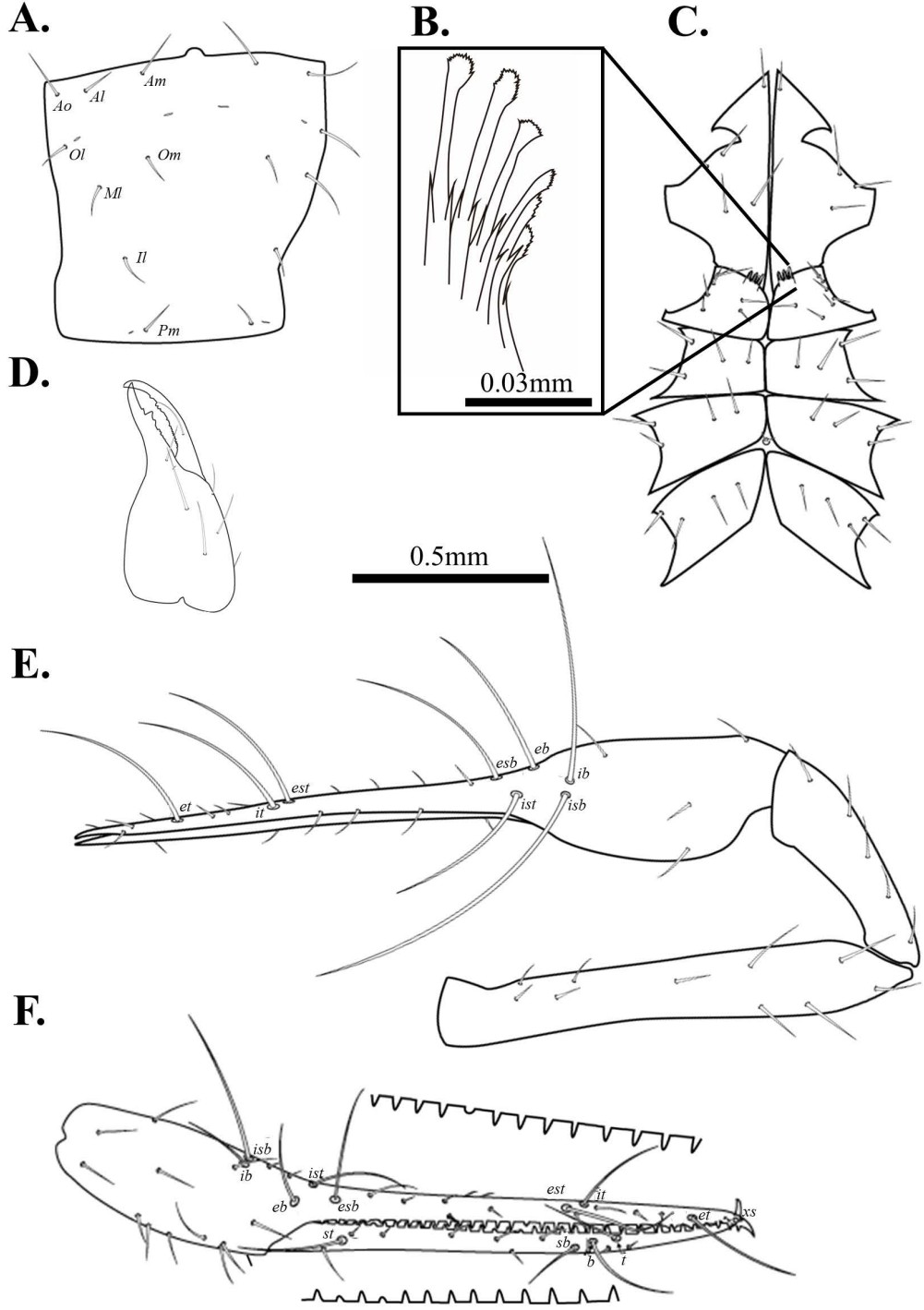

**Fig 8. Drawings of *Spelaeochthonius geumgulensis* sp. nov.** A. Carapace, dorsal view; B. Coxal spines; C. Leg Coxae I-IV; D. Chelicera, dorsal view; E. Right pedipalp, dorsal view; F. Right chela, lateral view. Scale bars: 0.5 mm (A, C–F); 0.05 mm (B).

Abdomen. Tergal chaetotaxy, 2: 2: 4: 4: 6: 6: 6: 7: 5: 4: 0: 0. Sternal chaetotaxy, 10: 13: 12: 12: 11: 9: 9: 7: 5: 2: 2.

Legs. Leg I: trochanter 1.50 times, femur 7.94 times, patella 5.26 times, tibia 4.35 times, tarsus 13.87 times longer than broad, leg IV: trochanter 1.42 times, femur+patella 4.27 times, tibia 6.15 times, metatarsus 4.33 times, tarsus 7.66 times longer than broad.

Dimensions (in mm). Body length 2.42. Pedipalp: trochanter 0.36/0.21, femur 1.16/0.21, patella 0.61/0.21, chela (with pedicel) 1.88/0.27, movable finger 1.17, hand 0.71/0.27. Chelicera: total 0.70/0.30, movable finger 0.36, hand 0.40/0.30. Carapace 0.74/0.65. Leg I: trochanter 0.24/0.16, femur 0.71/0.09, patella 0.43/0.08, tibia 0.28/0.06, tarsus 0.78/0.06. Leg IV: trochanter 0.27/0.19, femur+patella 1.07/0.25, tibia 0.81/0.13, metatarsus 0.38/0.09, tarsus 0.56/0.07.

### *Spelaeochthonius magwihalmigulensis* Jeong & Harms sp. nov.

**urn:lsid:zoobank.org:act:C098B292-D5E6-4D5C-97EC-D5461E787A55**

Figs 1–2, 9–10

**Type material.** Holotype. Female (NIBRIV0000924117 KOREA: Gangwo-Province: Gangneung-si, Okgye-myeon, Nakpung-ri, Magwihalmigul-cave; 37˚37' N 129˚0' E; 08 January, 2023; JH Oh leg).

**Etymology.** This species is named after the cave "Magwihalmigul" in the Gangneung-si, Gangwon-Province, where the species was sampled.

**Diagnosis.** This species is most similar to *Spelaeochthonius cheonsooi* by having a small pedipalpal femur (less than 1 mm), and a small pedipalpal patella (less than 0.4 mm). Both species differ in chelal length (1.49 mm *S. cheonsooi* vs. 1.16 mm in *S. magwihalmigulensis* **sp. nov.**) and the ratio between the movable chelal finger and the hand (1.67x, in *S. cheonsooi* vs. 2.38x in *S. magwihalmigulensis* **sp. nov.**).

**Description. Female, adult (holotype, Figs 9A–9B)**

Colour. Body pale-orange; chelicera darker than the body.

Chelicera (Fig 10D). Coarsely granulate; five setae on the cheliceral palm, one seta on the movable finger; nine marginal teeth on the movable finger, six on the fixed finger; one big tooth on the fixed finger; serrula exterior with 16 blades; rallum with 11 blades; one dorsal lyrifissure

Pedipalp (Figs 10E–10F). Trochanter 1.59 times, femur 5.51 times, patella 1.43 times, chela 6.78 times, hand 2 times longer than broad, movable finger 2.38 times longer than the hand. Fixed finger with eight trichobothria, movable finger with four trichobothria; *isb* and *ib* basally positioned on the dorsum of the fixed finger; *eb*, *ist*, and *esb* grouped, *ist* positioned

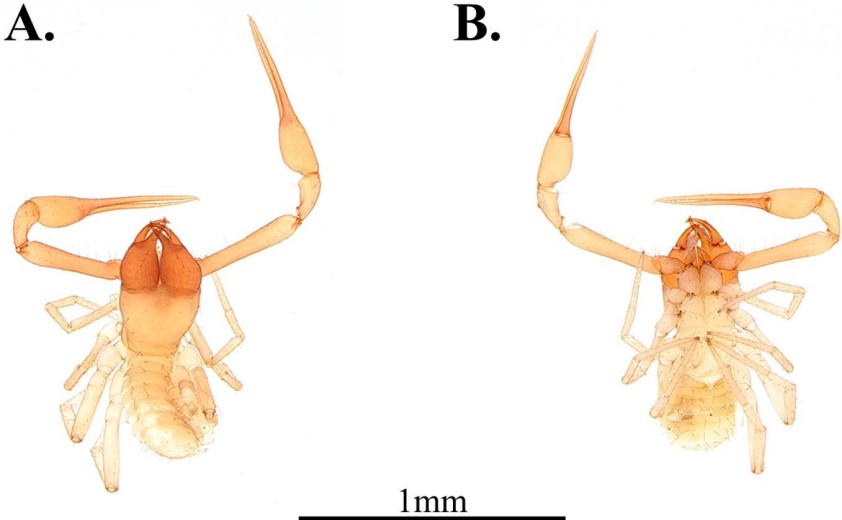

**A.**  **B.**

1mm

**Fig 9. Habitus of *Spelaeochthonius magwihalmigulensis* sp. nov.** A. Female, dorsal view; B. Female, ventral view. Scale bar: 1mm.

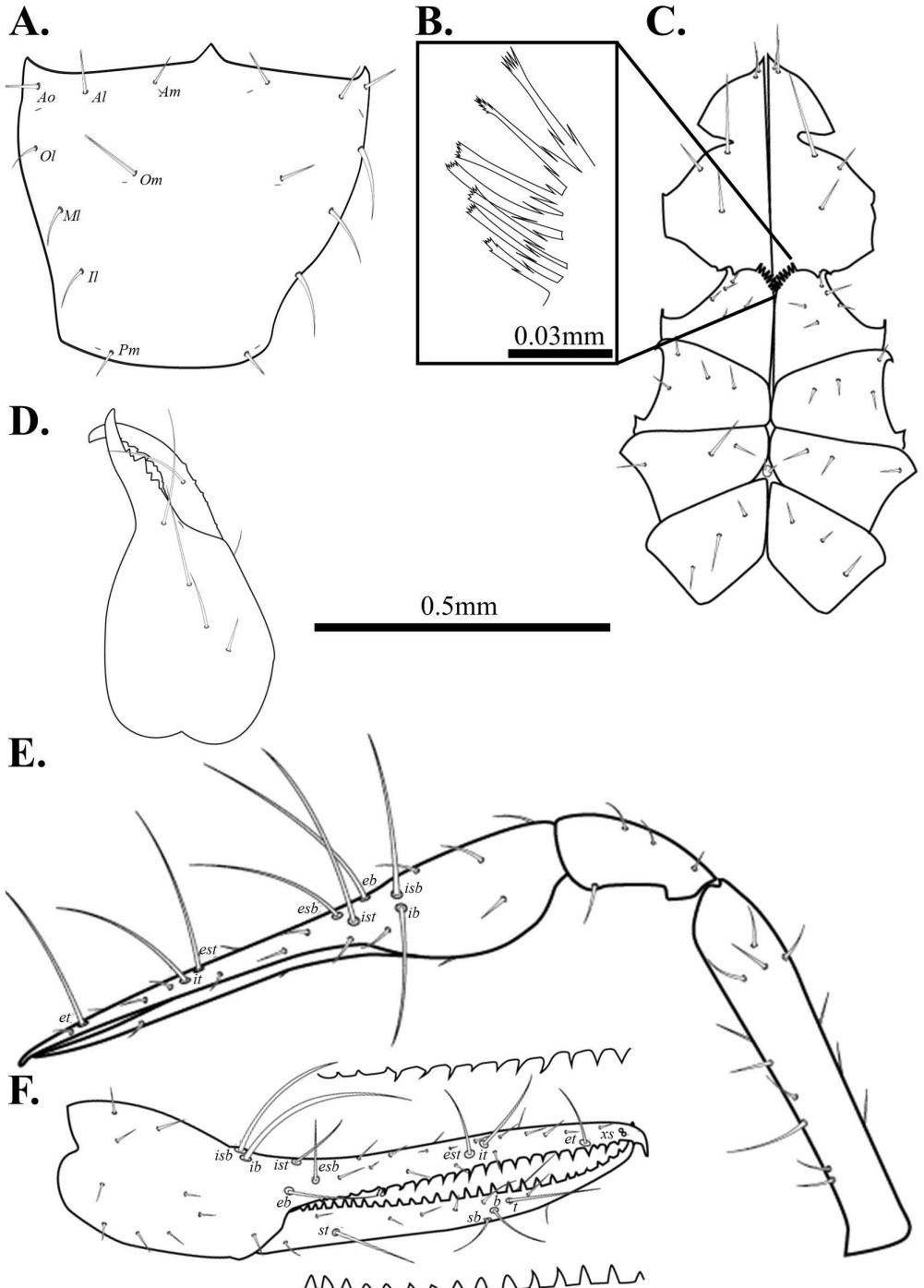

**Fig 10. Drawings of *Spelaeochthonius magwihalmigulensis* sp. nov.** A. Carapace, dorsal view; B. Coxal spines; C. Leg Coxae I–IV; D. Chelicera, dorsal view; E. Right pedipalp, dorsal view; F. Right chela, lateral view. Scale bars: 0.5 mm (A, C–F); 0.05 mm (B).

middle of *eb* and *esb*; *est* and *it* positioned on the middle of the fixed finger; *et* positioned between *xs* and *it*, but slightly closer the *xs*; *xs* located terminally on the fixed finger; *sb*, *b*, and *t* grouped, each apart about one areolar; *sb*, *b*, and *t* separately situated from *st*; *st* positioned basally on the movable finger. Fixed finger with 25, movable finger with 27 marginal teeth, spaced together; apical shape in anterior teeth, retrose and rounded teeth basally positioned on movable finger.

Cephalothorax (Figs 10A–10C). Carapace 0.90 times longer than the broad; lateral margin distinctly wider than the base; eyeless; epistome big and triangular; 16 setae on the carapace, three setae pairs on the anterior margin (*Ao*, *Al*, *Am*), setae pair on the lateral margin (*Ol*), three setae pairs on the medial margin (*Om*, *Ml*, *Il*), one setae pair on the posterior margin (*Pm*); seta short and acuminate. Two pairs of anterior lyrifissures, a pair of medial lyrifissures, a pair of posterior lyrifissures present. One acuminate seta on the manducatory process, three setae on the maxilla; coxal chaetotaxy 5: 4: 3: 3; coxal spines with eight blades, each blade tripartite; intercoxal tubercle distinct and bisetose.

Abdomen. Pleural membrane granulate; tergites undivided; tergal chaetotaxy 2: 2: 4: 4: 6: 6: 8: 8: 6: 4: 0: 0; sternites IV–V divided; sternal chaetotaxy 9: 10: 10: 10: 9: 10: 6: 6: 4: 2: 2.

Legs. Leg I: trochanter 1.56 times, femur 5.47 times, patella 1.43 times, tibia 4.6 times, tarsus 10.2 times longer than broad; leg IV: trochanter 1.78 times, femur+patella 5.13 times, tibia 4.19 times, metatarsus 3.33 times, tarsus 9.90 times longer than the hand. Arolium undivided. Pseudotactile seta situated basally on leg IV tarsus and metatarsus.

Female genitalia. Nine setae on the sternite II, ten setae on the sternite III.

Dimensions (in mm). Body length 1.26. Pedipalp: trochanter 0.24/0.15, femur 0.91/0.17, patella 0.33/0.23, chela 1.16/0.17, hand 0.34/0.17, movable finger 0.82. Chelicera: total 0.64/0.28, hand 0.32/0.28, movable finger 0.32. Leg I: trochanter 0.18/0.11, femur 0.48/0.09, patella 0.27/0.07, tibia 0.23/0.05, tarsus 0.51/0.05. Leg IV: trochanter 0.27/0.15, femur+patella 0.72/0.14, tibia 0.23/0.05, metatarsus 0.24/0.07, tarsus 0.52/0.05.

***Spelaeochthonius yamigulensis* Jeong & Harms sp. nov.**

**urn:lsid:zoobank.org:act:C783D03F-57BB-46A3-A16D-06551DB25AB3**

Figs 1–2, 11–12

**Type material.** Holotype. Male (NIBRIV0000924116 KOREA: Gangwon-Province: Jeongseon-gun, Najeon-ri, San 183−2, Yamigul-cave; 37˚25' N 128˚38' E; 13 Nov. 2022; K.-H. Jeong leg.)

**Etymology.** This species is named after the "Yamigul" cave in Jeongseon-gun, Gangwon-Province, where the species was collected.

**Diagnosis.** The species differs from other congeners by the number of carapacal setae. In contrast to other species within the genus, where seta *pl* is absent, in *S. yamigulensis* **sp. nov.** seta *pl* is present and there are 18 carapacal setae instead of 16 setae which is the standard for *Spelaeochthonius*.

**Description.** Male, adult (holotype, Fig 11A–11B)

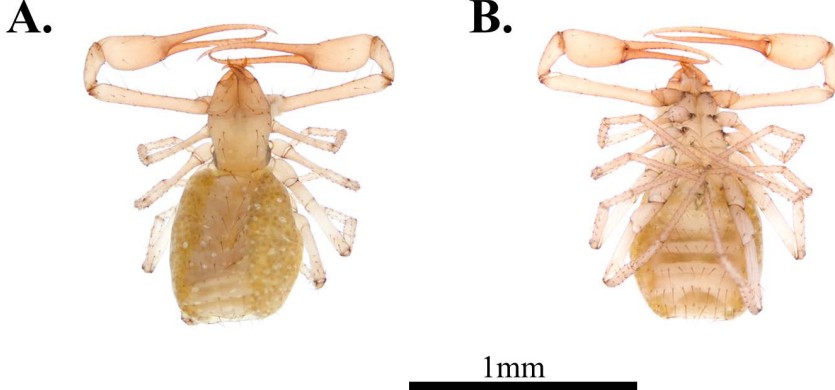

**Fig 11. Habitus of *Spelaeochthonius yamigulensis* sp. nov.** A. Male, dorsal view; B. Male, ventral view. Scale bar: 1mm.

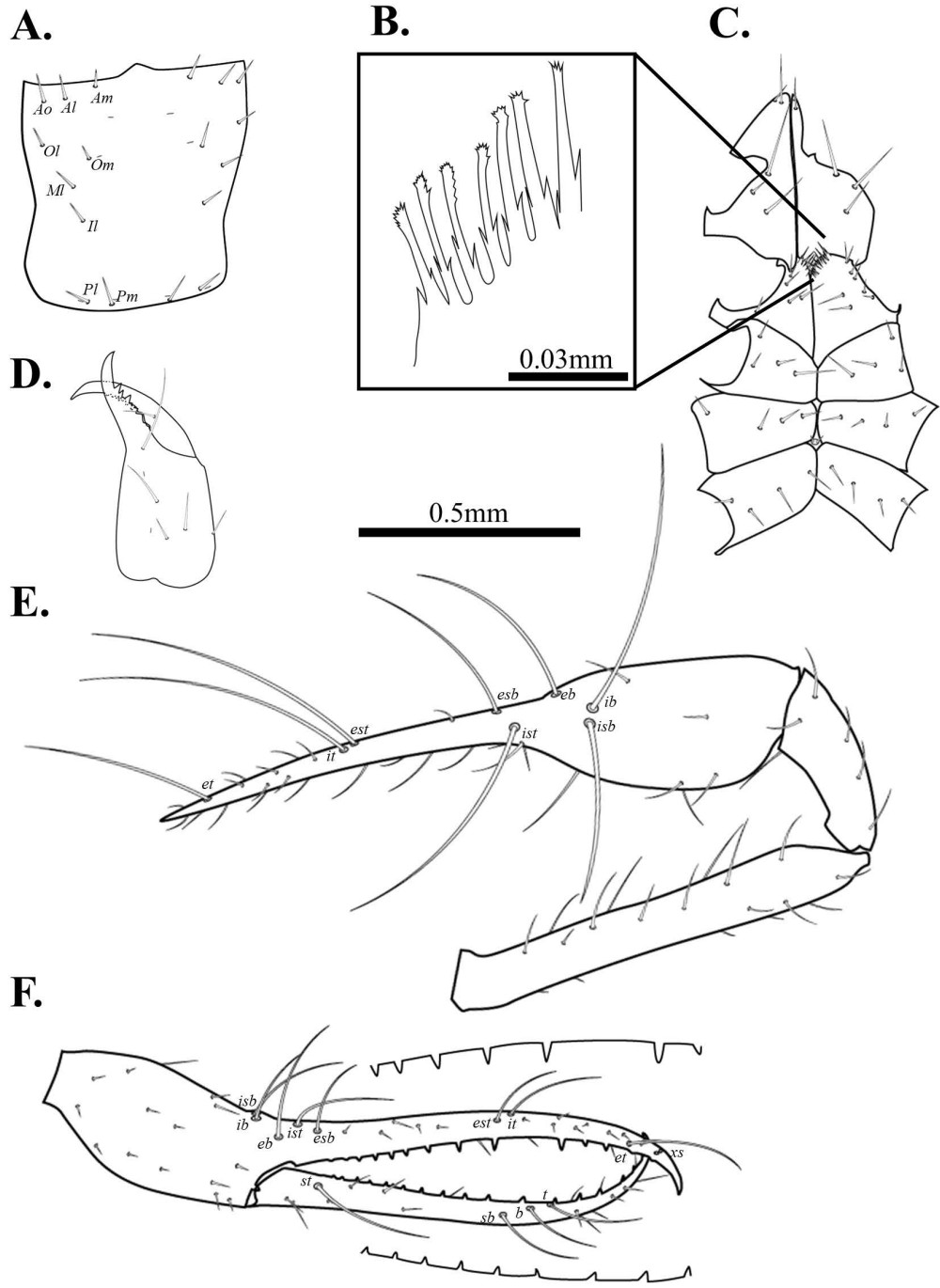

**Fig 12. Drawings of *Spelaeochthonius yamigulensis* sp. nov.** A. Carapace, dorsal view; B. Coxal spines; C. Coxa; D. Chelicera, dorsal view; E. Right pedipalp, dorsal view; F. Right chela, lateral view. Scale bars: 0.5 mm (A, C–F); 0.05 mm (B).

Colour. Body brown; chela pale-orange, darker at the tip.

Chelicera (Fig 12D). Cheliceral palm smooth; five setae on cheliceral palm, one on the movable finger; nine marginal teeth on the fixed finger, movable finger with nine marginal teeth; three big teeth on the terminal of the fixed finger; rallum with 10 blades; serrula exterior with 15 blades; one dorsal lyrifissure.

Pedipalp (Figs 12E–12F). Trochanter 1.65 times, femur 7.04 times, patella 2.45 times, chela 5.46 times, hand 2.19 times longer than broad, movable finger 1.49 times longer than the hand. Fixed finger with eight trichobothria, movable finger with four trichobothria; *isb* and *ib* basally positioned on the dorsum of fixed finger; *eb*, *ist*, and *esb* grouped, *ist* positioned slightly closer to *esb*; *est* and *it* positioned on the middle of fixed finger; *et* positioned near *xs*, about three areolar apart; *xs* located terminally on the fixed finger; *sb*, *b*, and *t* separately situated from *st*; *st* positioned basally on the movable finger. Fixed finger with 15, movable finger with 18 marginal teeth; apical teeth in both fingers.

Cephalothorax (Figs 12A–12C). Carapace 1.02 times longer than broad; anterior margin slightly wider than posterior margin; 18 setae on the carapace, three setae pairs on the anterior margin (*Ao*, *Al*, *Am*), one setae pair on the lateral margin (*Ol*), three setae pairs on the medial margin (*Om*, *Ml*, *Il*), two setae pairs on the posterior margin (*Pl*, *Pm*); seta acuminate. A pair of anterior lyrifissures, a pair of medial lyrifissures, two pairs of posterior lyrifissures present. Two acuminate setae on the manducatory process, three setae on the maxilla; coxal chaetotaxy 7: 4: 4: 4; coxal spines with eight blades; each blade tripartite; intercoxal tubercle distinct, two setae on the tubercle.

Abdomen. Pleural membrane granulate; tergites undivided; tergal chaetotaxy 2: 4: 4: 4: 5: 6: 6: 7: 7: 5: 4: 0: 0; sternite III divided, sternite IV partially divided; sternal chaetotaxy 9: 28: 14: 12: 15: 13: 11: 11: 8: 6: 2: 2.

Legs. Leg I: trochanter 1.48 times, femur 5.43 times, patella 5.17 times, tibia 3.93 times, tarsus 10.1 times longer than broad; leg IV: trochanter 1.33 times, femur+patella 4.86 times, tibia 6.31 times, tarsus 3.51 times, metatarsus 12.31 times longer than broad. Arolium undivided. Pseudotactile seta present on leg IV tarsus.

Male genitalia. Sternite II with nine setae; sternite III with 28 setae, 15 setae near the genital opening area. Sternite II with two lyrifissures; sternite III with six lyrifissures.

Dimensions (in mm). Body length 1.72. Pedipalp: trochanter 0.28/0.17, femur 1.04/0.15, patella 0.43/0.18, chela 1.56/0.29, hand 0.63/0.29, movable finger 0.93. Chelicera: total 0.59/0.23, hand 0.30/0.23, movable finger 0.29. Leg I: trochanter 0.22/0.15, femur 0.46/0.08, patella 0.38/0.07, tibia 0.28/0.07, tarsus 0.65/0.06. Leg IV: trochanter 0.20/0.15, femur+patella 0.80/0.17, tibia 0.59/0.09, metatarsus 0.26/0.07, tarsus 0.73/0.06.

**Remarks.** This species has a carapacal setation that differs from the diagnosis for *Spelaeochthonius* presented in You et al. (2022) [6]. The genus generally has 16 setae that are arranged s4s: 4: 2: 2: 2: 2 (Hou and Zhang 2024) but in *S. yamingulensis* sp. nov. seta *pl* is present and the pattern is 4: 2: 2: 2: 4. Similar derivations are known in other pseudotyrannochthoniid genera and an interesting analogy is *Centrochtonius anatonus* Harvey & Harms, 2022 that shows the same duplication and has 18 carapacal setae (*pl* present) whereas all other species have 16 setae (*pl* absent) (Harms & Harvey, 2022). Genetically but also in terms of coxal spine morphology, *S. yamingulensis* **sp. nov.** can clearly be placed in *Spelaeochthonius* and the number of posterior carapaceal setae seems to be somewhat variable within both genera.

## 4.2. Molecular analyses

The Maximum Likelihood (ML) phylogenetic analysis of the mtCOI dataset supported the species hypothesis that was derived from morphology and suggested the presence of at least seven species of *Spelaeochthonius* in South Korea (Fig 13). Although the bootstrap values were generally low at the basal nodes it appears that the Korean fauna of *Spelaeochthonius* is polyphyletic and that two species from Japan nested within the Korean fauna. Overall, the terminals can be classified into three genetic clades (A–C) that differ in terms of pairwise genetic distances that range between 13.7% to 19.8%. In clade B, the pairwise genetic distance between *S. cheonsooi* and *S. yamigulensis* **sp. nov.** is 10.8% and the pairwise genetic distance between *S. kobayashii* and *Spelaeochthonius* sp. equals 10%. In clade C, the genetic distance between *S. dentifer* and *S. magwihalmigulensis* **sp. nov.** is 6.8% and in *S. seungsookae* and *S. geumgulensis* **sp. nov.** is 15.6–16% (Table 1).

According to Cosgrove et al. (2016), > 10% COI divergences might be used for identifying putative species in Pseudoscorpiones at a molecular level [27]. However, the COI divergences between *S. dentifer* and *S. magwihalmigulensis* **sp. nov.** are lower (6.8%) but both species are clearly differentiated morphologically and differ in the number of teeth on both the fixed and movable chelal finger (25/27 in *S. magwihalmigulensis* **sp. nov.** vs 34/37 in *S. dentifer*), the ratio between the movable chelal finger and the hand (1.67 in *S. magwihalmigulensis* **sp. nov.** vs 2.17 in *S. dentifer*), and ratios of the chelal length (1.16 mm in *S. magwihalmigulensis* **sp. nov.** vs 1.60 mm in *S. dentifer*). Both species also originate from different karst systems that are approximately 47 km apart from each other and not interconnected.

## 5. Discussion

### 5.1. Species diversity of *Spelaeochthonius* in South Korea

*Spelaeochthonius* is a genus found exclusively in caves habitat of eastern Asia and the 15 described species in this genus are altogether blind, pale, and strongly troglobitic with the exception of the recently discovered *S. tuoliangensis* from China that retains residual eyes and shorter legs [28]. Collecting pseudoscorpions in caves is usually by chance and until recently, very few surveys have specifically targeted troglobitic pseudoscorpions in eastern Asia. The fauna of South Korea is no exception and two target surveys with limited spatial sampling [6, this study] have not only resulted in the discovery of new species, but also revealed that almost every cave system visited by the first author had a cave-adapted pseudoscorpion fauna that was completely unknown before. The Korean Peninsula has karstic formations sculptured on most major mountain ranges, such as Taebaek and Sobaek Mountain ranges, primarily in the eastern regions, and karstic studies indicate that there are more than 1,000 caves in South Korea alone [1]. The present study examined four cave populations, which exhibited distinct morphological and genetic differences from other cave populations. Based on these results, it is not difficult to perceive a hypothesis by which the pseudoscorpion fauna of Korean caves will be highly diverse

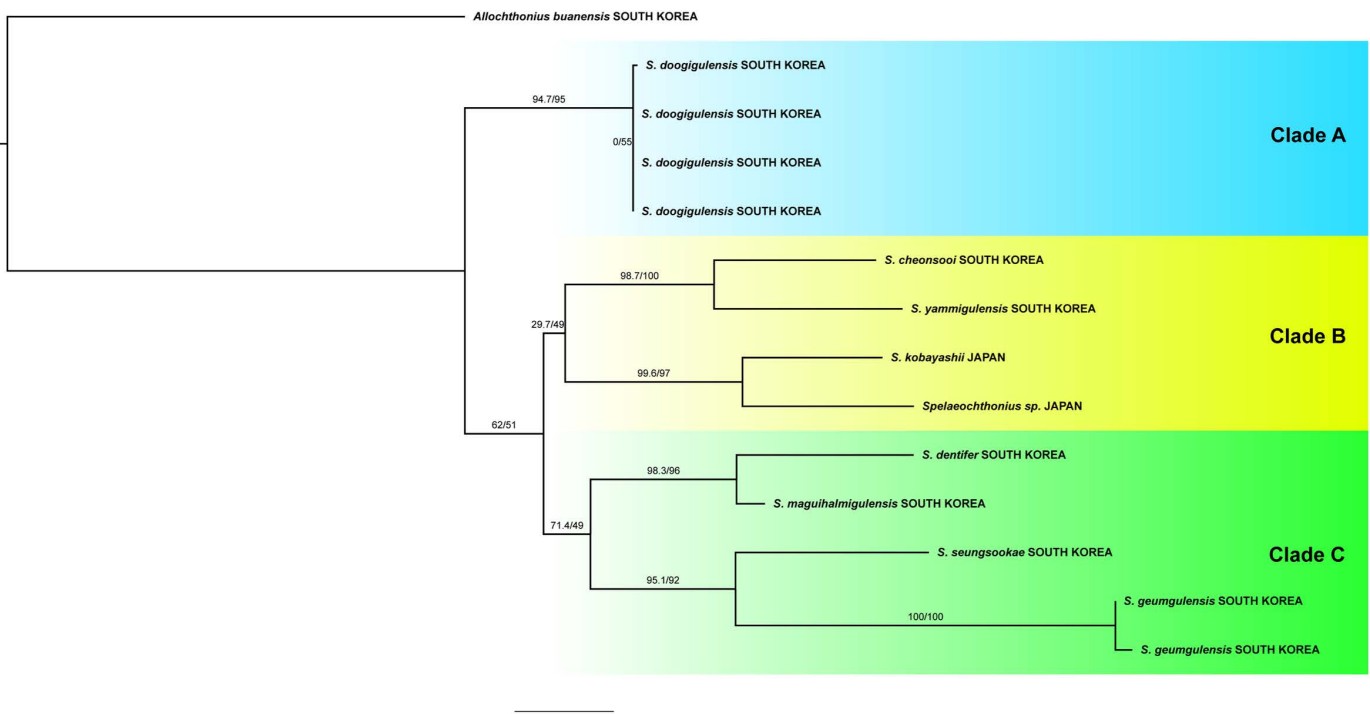

**Fig 13. Phylogenetic analyses of *Spelaeochthonius* based on maximum likelihood analysis with CO1.**

**Table 1. Genetic divergences within all species included in this study.**

**1) Intraspecific genetic divergence**

| Species (Comparison pairs) | Intraspecific divergence | | |
| --- | --- | --- | --- |
| | Maximum | Minimum | Average |
| *S. cheonsooi* | – | – | – |
| *S. dentifer* | – | – | – |
| *S. geumgulensis* **sp. n.** (n=2) | 0.6% | 0.6% | 0.6% |
| *S. dugigulensis* **sp. n.** (n=4) | 0.17% | 0 | 0.08% |
| *S. magwihalmigulensis* **sp. n.** | – | – | – |
| *S. seungsookae* | – | – | – |
| *S. yamigulensis* **sp. n.** | – | – | – |

2) Interspecific genetic divergence

| Species | *S. dentifer* | *S. kobayashii* | *S. seungsookae* | *S. geumgulensis* sp.n | *S. dugigulensis* sp.n. | *S. magwihalmigulensis* sp.n. | *S. yamigulensis* sp.n. |
| --- | --- | --- | --- | --- | --- | --- | --- |
| *Spelaeochthonius* sp. | 18.8% | 10.0% | 17.2% | 19.1–19.8% | 14.7–14.9% | 15.9% | 16.6% |
| *S. cheonsooi* | 18.1% | 16.4% | 18.7% | 18.8–19.2% | 16.2–16.4% | 14.9% | 10.8% |
| *S. dentifer* | | 16.3% | 14.7% | 18.9–19.3% | 16.3–16.4% | 6.8% | 17.4% |
| .*S. kobayashii* | | | 15.5% | 18.5–19.1% | 15.5% | 13.5% | 16.2% |
| .*S. seungsookae* | | | | 15.6–16.0% | 15% | 12.7% | 17.2% |
| *S. geumgulensis* sp.n. | | | | | 16.0–16.9% | 17.6–18.1% | 19.2–19.8% |
| *S. dugigulensis* sp.n. | | | | | | 13.7–13.9% | 15.9–16.0% |
| *S. magwihalmigulensis* sp.n. | | | | | | | 14.7% |

and comprise potentially dozens of new species [5,29]. Like similar cases in neighboring countries such as China, the pseudoscorpion biodiversity of Korean subterranean ecosystem is significantly higher than previously thought, underscoring the necessity for further research into these systems [11–13]. To summarize, discovering undescribed subterranean species is an important project for conserving and understanding the unveiled fauna of this dark ecosystem.

## 5.2. Endemism and biogeography

Due to the limited number of specimens that are usually collected during field work in caves, it is often challenging to infer intraspecific genetic divergences within cave species and total species ranges. The pseudoscorpion species in our present study are no different but the results demonstrate a clear distinction between the intraspecific and interspecific distances, which is consistent with the concept of the 'barcode gap' in molecular taxonomy suggesting that the variation found across species is higher than the genetic variability seen within a species [30]. Interspecific genetic distances in our study ranged from 6.8% to 19.8% but intraspecific distances observed in our study were always smaller than 0.6%. These results could be caused by under-sampling (e.g., only one cave in a complex karst system was sampled) but might also be real considering that genetic drift in cave populations often results in a reduction or loss of genetic variability [31–35]. It has also been observed frequently that gene flow between caves is often scarce and often results in the formation of numerous cryptic or closely related species [35–39]. The same has been recorded in non-phoretic pseudoscorpions [18,40] and our data are in line with such studies and suggest that scarce gene flow and genetic drift in cave populations could foster endemism in cave pseudoscorpions across the Korean Peninsula. This hypothesis could be explored further

using multi-gene phylogenetic studies involving nuclear molecular markers, or even genomics, and detailed specimen collections in multiple caves belonging to a single karst system. In the context of genetic data we reiterate once more that the COI pairwise divergences between *S. dentifer* and *S. maguihalgulensis* **sp. nov.** are somewhat lower than in previous studies [27]. However, morphological differences and distributional patterns support the hypothesis that they represent different species and other (nuclear) genetic markers might also recover these divergences. Also, the previous research was conducted with epigean species, so that the genetic distance could be different between the two examples.

Although only based on single-gene analyses, our study also suggests close affinities of the Southern Korean fauna of *Spelaeochthonius* to Japan; a result that was also recovered during a global phylogenetic study of Pseudotyrannochthoniidae using multiple genetic markers [41]. Two hypotheses can be evoked to explain the faunal affinities between the Korean Peninsula and the Japanese Archipelago. First, this genus might have been widely distributed throughout East Asia before the formation of the Sea of Japan. Alternatively, there might have been faunal exchange during the last glacial maximum when sea levels were lower and land connections existed between Japan and Korea [42,43]. More extensive field sampling and comprehensive phylogenetic analyses are necessary to test these hypotheses rigorously but the data presented in Harms et al. (2024) suggest that these relationships are old and certainly predating the last glacial periods [41].

### 5.3. Caves as a major contributor to Korean diversity and conservation values

Caves in Korea support diverse communities of invertebrates, including pseudoscorpions. According to Kim et al. (2004), at least 260 species of invertebrates that belong to four phyla, 31 orders and 94 families are known from caves in Korea [44]. There are 97 species of spiders alone and the known cave diversity represents 14% of all spider species found in South Korea [45] which is a substantial number considering the poor state of cave research in Korea. Of the 33 pseudoscorpion species now recorded in Korea, eight species (24%) are troglobites and this number will surely rise in the future.

Cave pollution and the destruction of caves are unfortunately common problems in South Korea, with sometimes devastating consequences. For example, the carabid beetle *Coreoblemus parvicollis* Uéno, 1969 was known only from Cheongpungpunghyeol Cave and last sighted in 1967 before the completion of the Chungju-dam. The species is now listed as extinct [46]. According to Kim & Kim (2007), several lava tubes on Jeju Island are under severe destruction threat because of road construction and pollution by tourists [47]. It is clear that the emerging karst biodiversity in Korea is underrepresented in practical nature conservation and presently underappreciated and not at all protected by conservation legislation. Target surveys in combination with taxonomy and distribution modelling for cave fauna in South Korea are suggested to improve both knowledge and practical conservation of cave fauna across the country.

A small number of specimens were included in this study (four specimens for *S. dugigulensis* **sp. nov.**, two specimens for *S. geumgulensis* **sp. nov.**, and one specimen each for *S. magwihalmigulensis* **sp. nov.** and *S. yamigulensis* **sp. nov.**). Because of the rarity of subterranean species and the danger of accessing caves, it is common to use a limited number of specimens for descriptions [13,48]. And, considering the cryptic diversity of subterranean pseudoscorpion fauna in South Korea, the description of unknown species is an urgent task. Taxonomic knowledge and biological background are critical for conservation efforts, helping to prevent 'dark extinction' of these cryptic animals [49–51].

### Acknowledgments

KHJ would like to thank Yong-Gun Choi (Korean Society of Cave Environment) for guiding caves, and Dongyoung Kim and Jonghwa Oh (Seoul National University) for assistance in the field.

### Author contributions

**Conceptualization:** Kyung-Hoon Jeong, Danilo Harms, Jung-sun Yoo, Sora Kim.

**Data curation:** Kyung-Hoon Jeong, Sora Kim.

**Formal analysis:** Kyung-Hoon Jeong.

**Funding acquisition:** Jung-sun Yoo, Sora Kim.

**Investigation:** Kyung-Hoon Jeong, Danilo Harms, Jung-sun Yoo, Sora Kim.

**Methodology:** Kyung-Hoon Jeong, Danilo Harms.

**Project administration:** Danilo Harms, Jung-sun Yoo, Sora Kim.

**Supervision:** Danilo Harms, Jung-sun Yoo, Sora Kim.

**Visualization:** Kyung-Hoon Jeong, Danilo Harms, Sora Kim.

**Writing – original draft:** Kyung-Hoon Jeong.

**Writing – review & editing:** Kyung-Hoon Jeong, Danilo Harms, Jung-sun Yoo, Sora Kim.

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
