## [Decision Letter · Decision Letter 0]

Dear Dr. Kim

Thank you for submitting your manuscript to PLOS ONE. After careful consideration, we feel that it has merit but does not fully meet PLOS ONE’s publication criteria as it currently stands. Therefore, we invite you to submit a revised version of the manuscript that addresses the points raised during the review process.

Dear authors,

I have received the comments from both reviewers, and they have suggested **minor revisions**for your manuscript. Please review their feedback carefully and make the necessary revisions accordingly.

Looking forward to hearing from you.

All the best,

Mostafa

We look forward to receiving your revised manuscript.

Kind regards,

Mostafa Ghafouri Moghaddam, Ph.D

Academic Editor

PLOS ONE

 [This work was supported by a grant from the National Institute of Biological Resources (NIBR), funded by the Ministry of Environment (MOE) of the Republic of Korea (NIBR202502102).].

6. Please include a separate caption for each figure in your manuscript.

7. We note that Figures 1 and 2 in your submission contain [map/satellite] images which may be copyrighted. All PLOS content is published under the Creative Commons Attribution License (CC BY 4.0), which means that the manuscript, images, and Supporting Information files will be freely available online, and any third party is permitted to access, download, copy, distribute, and use these materials in any way, even commercially, with proper attribution. For these reasons, we cannot publish previously copyrighted maps or satellite images created using proprietary data, such as Google software (Google Maps, Street View, and Earth). For more information, see our copyright guidelines: http://journals.plos.org/plosone/s/licenses-and-copyright.

1. You may seek permission from the original copyright holder of Figures 1 and 2 to publish the content specifically under the CC BY 4.0 license. 

Additional Editor Comments:

Dear authors,

I have received the comments from both reviewers, and they have suggested minor revisions for your manuscript. Please review their feedback carefully and make the necessary revisions accordingly.

Looking forward to hearing from you.

All the best,

Mostafa

Reviewers' comments:

Reviewer's Responses to Questions

**Comments to the Author**

1. Is the manuscript technically sound, and do the data support the conclusions?

Reviewer #1: Yes

Reviewer #2: Yes

2. Has the statistical analysis been performed appropriately and rigorously?

Reviewer #1: Yes

Reviewer #2: Yes

3. Have the authors made all data underlying the findings in their manuscript fully available?

Reviewer #1: Yes

Reviewer #2: Yes

4. Is the manuscript presented in an intelligible fashion and written in standard English?

Reviewer #1: Yes

Reviewer #2: Yes

Reviewer #1: This is a very nice study which describes four new species of Spelaeochthonius from caves in South Korea. The importance of the present work is that the new species are supported by both morpholodical and molecular date. It is certainly worth publishing in PLOS ONE. The text and figures are very good throughout, and I have only MINOR REVISIONS to suggest, see attached DOCX.

Reviewer #2: The manuscript has been well written and is an important contribution towards the Korean Karst system. However, I notice some typos here and there, some punctuation error, which I have annotated in the pdf attached. Apart from these, the manuscript is good to go.

**Do you want your identity to be public for this peer review?** For information about this choice, including consent withdrawal, please see our Privacy Policy

Reviewer #1: **Yes: ** Feng Zhang

Reviewer #2: **Yes: ** Jithin Johnson

---

## [Author Response · Author response to Decision Letter 1]

28 Apr 2025

Editors

Comment #1 (Manuscript style): We have revised the manuscript to fully comply with the journal’s formatting and style requirements.

Comment #2 (Sampling methods): We have added the following sentence to the Materials and Methods section for clarification, "Field surveys were conducted with the assistance of the Korean Society of Cave Environment or conducted in publicly accessible areas."

Comment #3 (Funding statement): The Funding Statement has been updated to:

"This work was supported by a grant from the National Institute of Biological Resources (NIBR), funded by the Ministry of Environment (MOE) of the Republic of Korea (NIBR202502102). There was no additional external funding received for this study."

Comment #4 (ORCID ID): We have updated the ORCID ID of the corresponding author.

Comment #5 (Figures): Figures 1 and 2 were generated using datasets from the Global Multi-resolution Terrain Elevation Data 2010 (GMTED2010), provided by the United States Geological Survey (USGS) and the National Geospatial-Intelligence Agency (NGA), and vector data from the Natural Earth project. Both datasets are public domain resources that permit free use, including for commercial purposes, without the need for attribution. Therefore, the maps comply with the Creative Commons Attribution License (CC BY 4.0) required for publication. In methods section, we added the statements including those datasets are comply with CC BY 4.0.

Reviewer #1, Dr. Jithin Johnson

Thank you for providing valuable advise on the manuscript. We changed every typos you pointed out, and here are some comments and responses.

Comments #1 (typos): We changed every typos that reviewer pointed out. Thank you for your sharped suggestion.

Comments #2 (problems in the “References”): Thank you for your suggestion. We revised everything you mentioned.

Reviewer #2, Dr. Feng Zhang

Comments #1 (problems in the “References”): We added every absent references. Also, we deleted the reference “Lee et al., 2008”.

Comments #2 (problems in the “Spelaeochthonius dugigulensis”): Thank you for your detailed revision. Firstly, we recalculated the ratios of appendages and revised them accurately. Also, we checked the differences between S. dugigulensis sp. nov. and S. akiyoshiensis and changed the length to L/W ratios of chela, and deleted the number of coxal spines.

Comments #3 (problems in the “Figure 1”): We added distributions of the missing species, “Spelaeochthonius tuoliangensis and S. huanglaoensis”

---

## [Editor Report · Decision Letter 1]

Four new species of dragon pseudoscorpions (Pseudoscorpiones: Pseudotyrannochthoniidae: Spelaeochthonius) from caves in South Korea revealed by integrative taxonomy

PONE-D-25-09020R1

Dear Dr. Sora Kim,

We’re pleased to inform you that your manuscript has been judged scientifically suitable for publication and will be formally accepted for publication once it meets all outstanding technical requirements.

Kind regards,

Mostafa Ghafouri Moghaddam, Ph.D

Academic Editor

PLOS ONE
---

## [Editor Report · Acceptance letter]

PONE-D-25-09020R1

PLOS ONE

Dear Dr. Kim,

I'm pleased to inform you that your manuscript has been deemed suitable for publication in PLOS ONE. Congratulations! Your manuscript is now being handed over to our production team.

Kind regards,

on behalf of

Dr Mostafa Ghafouri Moghaddam

Academic Editor

PLOS ONE